# Molecular Biomarkers in Idiopathic Pulmonary Fibrosis: State of the Art and Future Directions

**DOI:** 10.3390/ijms22126255

**Published:** 2021-06-10

**Authors:** Anna Stainer, Paola Faverio, Sara Busnelli, Martina Catalano, Matteo Della Zoppa, Almerico Marruchella, Alberto Pesci, Fabrizio Luppi

**Affiliations:** 1Department of Medicine and Surgery, University of Milano Bicocca, 20126 Milano, Italy; annetta.stainer@gmail.com (A.S.); paola.faverio@unimib.it (P.F.); m.catalano17@campus.unimib.it (M.C.); alberto.pesci@unimib.it (A.P.); 2Respiratory Unit, San Gerardo Hospital, 20900 Monza, Italy; busnelli.sara@gmail.com (S.B.); a.marruchella@asst-monza.it (A.M.); 3Pulmonology Unit, Fondazione IRCCS Policlinico San Matteo, 27100 Pavia, Italy; matteo.dellazoppa@gmail.com

**Keywords:** idiopathic pulmonary fibrosis, biomarker, diagnosis, prediction

## Abstract

Idiopathic pulmonary fibrosis (IPF), the most lethal form of interstitial pneumonia of unknown cause, is associated with a specific radiological and histopathological pattern (the so-called “usual interstitial pneumonia” pattern) and has a median survival estimated to be between 3 and 5 years after diagnosis. However, evidence shows that IPF has different clinical phenotypes, which are characterized by a variable disease course over time. At present, the natural history of IPF is unpredictable for individual patients, although some genetic factors and circulating biomarkers have been associated with different prognoses. Since in its early stages, IPF may be asymptomatic, leading to a delayed diagnosis. Two drugs, pirfenidone and nintedanib, have been shown to modify the disease course by slowing down the decline in lung function. It is also known that 5–10% of the IPF patients may be affected by episodes of acute and often fatal decline. The acute worsening of disease is sometimes attributed to identifiable conditions, such as pneumonia or heart failure; but many of these events occur without an identifiable cause. These idiopathic acute worsenings are termed acute exacerbations of IPF. To date, clinical biomarkers, diagnostic, prognostic, and theranostic, are not well characterized. However, they could become useful tools helping facilitate diagnoses, monitoring disease progression and treatment efficacy. The aim of this review is to cover molecular mechanisms underlying IPF and research into new clinical biomarkers, to be utilized in diagnosis and prognosis, even in patients treated with antifibrotic drugs.

## 1. Introduction

Idiopathic pulmonary fibrosis (IPF) is a chronic, progressive lung disease [1]. The epidemiology of this disease is not uniform due to data collection methods and classification terms variability among different studies. However, throughout Europe and North America, an incidence between 2.8 and 19 cases per 100,000 people per year has been reported [2,3,4]. IPF primary affects men, older than 50 years (median age at diagnosis is about 65 years) [5,6,7]. The disease course is variable, due to different clinical phenotypes [8,9]. However, the median survival time from diagnosis is 2–4 years [10]. Since in its early stages, IPF may be asymptomatic, leading to a delayed diagnosis. When present, the most frequent symptoms are progressive dyspnoea and cough. The IPF diagnosis is based on the identification of the usual interstitial pneumonia (UIP) pattern, both on histological samples or radiological images, and the exclusion of other known causes of pulmonary fibrosis. Frequently the diagnosis is complex, requiring a multidisciplinary evaluation as recommended by international guidelines [11,12]. At present, two drugs, nintedanib and pirfenidone, which slow the progression of the disease and improve prognosis, are approved for the treatment of IPF [12].

The management of IPF is currently based on clinical data, such as symptoms, lung function tests, and radio-histological patterns, due to a lack of reliable molecular markers. However, the identification of clinical biomarkers, diagnostic, prognostic, and theranostic would allow an evaluation based on underlying pathobiological mechanism of disease, leading to adequate phenotyping of patients in terms of diagnosis, prognosis, and response to therapy. This review aims to cover molecular mechanisms underlying IPF and research into new clinical biomarkers, to be utilized in diagnosis and prognosis, even in patients treated with antifibrotic drugs.

## 2. Definition of a Biomarker

Biomarkers are defined as “characteristics that are objectively measured and evaluated as an indicator of normal biologic processes, pathogenic processes or pharmacologic responses to a therapeutic intervention” [13]. At any time during the evaluation of patients affected by a disease, biomarkers can be considered useful tools. Predisposition biomarkers could identify people at risk for eventually develop a disease, diagnostic and prognostic biomarkers integrate the diagnostic process and theranostic biomarkers are a reliable measure of efficacy and safety during treatment. Moreover, biomarkers are frequently used as a surrogate endpoint in clinical trials helping predict clinical benefit based on epidemiologic, therapeutic, pathophysiologic, or other scientific evidence [13]. Currently, 2018 American Thoracic Society (ATS), European Respiratory Society (ERS), Japanese Respiratory Society (JRS), American Latin Thoracic Association (ALAT) strongly recommends not to measure any serum biomarker for the purpose of distinguishing IPF from other interstitial lung diseases (ILD) in patients with newly detected ILD of apparently unknown cause who are clinically suspected of having IPF. Moreover, no guidelines or official statement on prognostic and theranostic biomarkers are available.

A good-quality biomarker should be reproducible, very sensitive, specific, and accurate. It should be validated in large multicentric trials and heterogeneous populations. Moreover, to be used on a large scale, it should be easily available and accessible. Biomarkers detectable on peripheral blood, exhaled breath condensate or broncho-alveolar lavage (BAL) offer an increased range of applications compared with a transbronchial or surgical lung biopsy. Finally, the cost-effectiveness ratio should be acceptable [14].

## 3. Molecular Biomarkers in IPF

The development of new molecular biomarkers for IPF is based on two different approaches. The hypothesis-driven method selects new candidate biomarkers a priori based on previous evidence about the disease. In contrast, the unbiased approach utilizes methods from systems biology to screen a large number of candidate biomarkers for their association with the disease. Although the former has the advantage of a strong rationale but lacks efficiency, the latter is more efficient but also burdened by the risk of false discovery [14]. 

Historically, IPF was considered a chronic inflammatory disorder, gradually leading to fibrosis. However, anti-inflammatory, and immunosuppressive therapy have shown to be ineffective and associated with increased mortality [15]. Up to date, IPF is described as characterized by the interaction of multiple genetic and environmental risk factors, with local micro-injuries to ageing alveolar epithelium. As a consequence, different process such as aberrant epithelial–fibroblast communication, the induction of myofibroblasts and the accumulation of extracellular matrix, lead to remodeling of lung interstitium [16]. Consequently, the most promising biomarkers in IPF are related to alveolar epithelial cell dysfunction, immune dysregulation, fibroproliferation, fibrogenesis, and extracellular matrix remodeling [14].

The main biomarkers analyzed in this review and their possible applications are resumed in Table 1 and Figure 1.

## 4. Predisposition Biomarkers

Mechanism or biological pathways linked to disease predisposition are reflected by predisposition biomarkers. They should provide information through inexpensive and non-invasive sampling with high sensitivity, specificity, and predictive value. Using predisposition biomarker, a patient could be address to informative counselling, preventive measures, and early therapy [14].

Surfactant proteins are secreted in surfactant by type II alveolar epithelial cells (AEC). They are encoded by SFTPA, SFTPB, SFTPC, and SFTPD genes [17]. Among surfactant proteins variants of surfactant protein C (SP-C) [18,19,20,21], surfactant protein A2 (SP-A2) [22,23,24,25] and surfactant protein A1 (SP-A1) [26] have been associated to familiar pulmonary fibrosis, while they are rare in sporadic IPF [19]. As surfactant protein levels can be measured in bronchoalveolar lavage fluid (BALF) and in blood, they could have a role in identifying at risk individuals in families with pulmonary fibrosis.

A common single nucleotide polymorphism (SNP) in the putative promoter region of the mucin 5B (MUC5B) gene (rs35705950), which encoded a glycosylated macromolecular component of mucus, has been associated with familiar pulmonary fibrosis and sporadic pulmonary fibrosis [27,28,29,30,31,32,33,34]. A recent meta-analysis confirmed that the minor T allele is significantly associated with an increased risk of IPF compared with the G allele in an allele dose-dependent manner [35]. However, MUC5B (rs35705950) has been found in 9% of people with interstitial lung abnormalities (ILA), a prevalence way higher than the rate reported for IPF [36]. Although MUC5B promoter polymorphism is a promising predisposition biomarker, it is neither necessary nor sufficient to cause the disease and understanding its role in IPF pathogenesis together with other genetic or environmental factors remain an unmet need.

The telomerase complex is involved in protection of chromosomes from loss of material, catalyzing the addition of repeated DNA sequences in the telomere region [37]. Several proteins contribute to the correct activity of the telomerase complex, including telomerase reverse transcriptase (TERT), dyskerin, telomere binding protein (TIN2), interaction with the telomerase repeat binding factor (TERF1), and the telomerase RNA component (TERC). Moreover, several other proteins contribute to the regulation of telomerase complex [37]. Several variants of the telomerase complex and its regulatory proteins have been associated to pulmonary fibrosis, especially familiar forms [38]. Although the telomerase complex can be evaluated on blood cells (granulocytes or monocytes), and common in IPF patients compared with age matched controls [39], it is globally rare in sporadic IPF and not specific since it has been associated with risk of developing chronic obstructive pulmonary disease (COPD) too [40]. 

Although the relationship between IPF and immunity is controversial, several components of the immune system have been evaluated as predisposition biomarkers in IPF. Toll-like receptors (TLR), fundamental components of innate immunity, have been implicated in the pathogenesis of IPF. TLR-2 mRNA is overexpressed in IPF patients and has shown pro-fibrotic features in mice. TLR-3 has shown antifibrotic features both in human and mice through downregulation of transforming growth factor beta 1 (TGFB1) and upregulation of prostaglandin E2 (PGE2). The loss-of-function variants (L412F) of TLR-3 lead to enhanced fibrotic responses. Some evidence, mainly in animal models, has been shown also for TLR2/4, TLR9, and TLR4 [41,42,43,44]. Another candidate gene for IPF is ELMOD-2, expressed in alveolar macrophages and type II AECs. A genome wide scan in 6 families with familiar pulmonary fibrosis in Finland showed reduced levels of mRNA expression of ELMOD-2 in IPF patients compared with healthy controls [45].

Predisposition biomarkers help understanding the pathogenesis of IPF and predicting the predisposition and prognosis of the disease. However, to date, none of these biomarkers is completely specific and sensitive for the diagnosis of sporadic pulmonary fibrosis, nor validated in clinical use. In familiar cases of pulmonary fibrosis, a consult with a geneticist and a screening for the most common biomarkers should be proposed to patients. 

## 5. Diagnostic Biomarkers

Diagnostic biomarkers should reflect the mechanism or biological pathways that distinct IPF from the other ILDs. They should be easy to evaluate and reproducible, such as blood, urine, BALF derived or imaging-based biomarkers. Ideally, they should improve diagnosis, reduce the risk of diagnostic tests, reduce the number of unclassifiable cases, helping discriminate IPF from other ILDs accurately [14]. Several blood proteins have shown some evidence in terms of diagnostic process, however, the use of none of them is recommended by guidelines for the diagnosis of IPF [11].

The detection of surfactant proteins in serum of patients with pulmonary diseases reflects an injury of the alveolar epithelial barrier. SP-A and D have been studied as diagnostic markers in IPF. Although BALF levels of surfactant proteins are reduced both in IPF and other ILDs compared with healthy controls [46,47], serum levels appear to be increased [47,48,49]. Wang et al. conduced a meta-analysis to evaluate the use of SP-A and SP-D for differential diagnosis of IPF. SP-A serum levels appear to be significantly higher in patients with IPF than in patients with non-IPF ILDs, pulmonary infection, and healthy controls, while no differences are found in SP-D serum levels in IPF versus non-IPF ILD patients, although higher than those in pulmonary infection and healthy controls [50]. Recently, SP-B precursor, C-pro-SP-B, has been studied as a new biomarker in serum of patients with different chronic lung diseases including ILDs. The highest serum levels of C-pro-SP-B were detected in the serum of IPF patients being able to differentiate IPF patients from patients with all other pulmonary diseases [51]. Moreover, SP-D levels were significantly elevated in acute exacerbation of IPF compared with stable IPF [52].

Krebs von den Lungen-6 (KL6)/mucin 1 (MUC1) is a glycoprotein expressed on the extracellular surface of type II AECs and bronchiolar epithelial cells in the lung largely studied in ILDs due to its overexpression in affected lung and regenerating type II AECs [53,54]. KL-6 is increased in serum of several ILDs including IPF [47,49,53,55,56,57] In one study, KL-6 levels in BALF seems to be a specific diagnostic marker in IPF compared with other ILDs [58] while Bennet et al. proved that higher levels of BALF KL-6 are related to a more severe and extended disease [56]. However, since KL-6 reflects AECs damage, it is not specific enough to distinguish IPF from the other ILDs nor alone neither as a part of composite index. However, it could facilitate stratification of severity [55,56,57].

Circulating caspase-cleaved cytokeratin-18 (cCK-18) is the cleaved fragment of cytokeratin-18 (CK-18), a cytoskeletal protein found in AECs. Since cCK-18 is produced during apoptosis in response to stress, it has been evaluated as a diagnostic and prognostic marker in one study: cCK18 was significantly elevated in the serum of IPF patients compared with normal controls and patients with other ILDs although it was not associated with prognosis [59].

TLRs have been studied widely for their implication in IPF pathogenesis and predisposition. However, some evidence has highlighted a possible role in diagnosis. Higher levels of TLR in BALF of IPF patients, particularly TRL-7 has been noted. In the same study, TLR also showed different profiles of expression in fibrotic and granulomatous disorders [60].

Metalloproteases (MMP) are another class of proteins widely studied for their role in in the aberrant fibrotic process, but the mechanisms are not completely understood and characterized as it seems they are implied both in deleterious and beneficial effects on the fibrotic process [61] They are a family of zinc-dependent matrixins that participate in extracellular matrix degradation but also in processing and cleaving of different bioactive mediators [62]. In particular, MMP-1 is upregulated in IPF patients compared with controls, and higher levels of MMP-1 has been shown in BALF and in plasma of IPF patients [63,64]. MMP-7 is also upregulated in IPF, with higher serum and BALF levels in patients compared with healthy controls [14,65]. Recently, Bauer et al. analyzed samples from the Bosentan Use in Interstitial Lung Disease (BUILD)-3 trial dosing MMP-7 among other biomarkers. MP-7 protein levels were elevated in IPF patients compared with healthy controls, and MMP-7 levels also increased over time [66]. Although MMP-7 alone is not sufficiently specific to distinguish IPF from other ILDs, if evaluated with other markers of fibrosis it could help differentiate IPF from other ILDs with good accuracy [64,67].

Osteopontin (OPN) is a phosphorylated glycoprotein that work as a mediator of inflammation and wound healing [68]. OPN is overexpressed in IPF lung [68,69] and seems its profibrotic role seems to be related to its ability to enhance fibroblasts migration by cooperating with chemoattractant interleukin 6 (IL-6) [69]. Moreover, OPN also induces upregulation of MMP-7 [70]. Although OPN is increased in serum and BALF of IPF patients [71,72], it is not specific in differentiating IPF from other ILDs [72]. However, as part of a composite index it helps improving diagnostic confidence [67].

## 6. Prognostic Biomarkers

Prognostic biomarkers should contribute to quantitative assessment of mechanism or biological pathways relevant to disease progression. They should be repeatable over time without significant risk for patients, such as blood and urine-based biomarkers. Moreover, they should have a low intra-patient, inter-test variability with calibration and discrimination values clearly established. They could be integrated in multiparametric models in order to improve prognostic counselling [14]. Potentially useful prognostic biomarkers have been identified both in genomic variants and blood proteins. With regard for genomic variants mutation in the telomerase complex and MUC5B have been studied as possible prognostic biomarkers. Telomere length has been associated to a worse survival [73] and transplant free survival [74] in IPF patients. Although MUC5B promoter variant (rs35705950) has been associated to an increased risk of developing pulmonary fibrosis, its role in predicting survival is contradictory [75,76,77,78]. Serum levels of SP-A and SP-D have been associated with reduced survival in IPF [46,48,79,80,81]. The variant in the TOLLIP gene, encoding for an adapter protein, is associated with a worse survival and more rapid disease progression possibly helping stratification at baseline of IPF patients [78].

Several studies highlighted a relationship between elevated values of KL-6 and mortality or progression in IPF [57,82,83,84,85] although these data have not been confirmed in other studies [86,87]. On the other hand, serial measurements of serum KL-6 concentrations resulted a risk factor for progressive disease and worse prognosis [57,88]. With regard for AE of IPF, serum values of KL-6 resulted higher compared with stable patients and higher values are predictor of onset of AE [52,89,90]. A recent systematic review and meta-analysis suggest that increased values of KL-6 in IPF is a predictor of AE risk, while it seems not to be related with mortality [87].

With regard for immune mediators, the SNP in theTLR3 (L412F) has been associated to increased mortality and accelerated progression in independent cohort [91,92]. Alpha-defensins, small antimicrobial proteins secreted by neutrophils and epithelial cells, has been proposed as a biomarker of AE of IPF. In fact, although alpha-defensins are upregulated in IPF lung, higher levels are detectable in patients with AE of IPF. Moreover, alpha-defensin serum levels were increased in AE IPF compared with stable IPF suggesting their use as biomarkers for AEs [63].

MMPs, in particular MMP-7, have been studied not only as diagnostic biomarkers but they can be useful tools in predicting prognosis and transplant free survival in IPF patients [14,66]. MMP-7 has also been evaluated in several studies in association with other markers of IPF for its diagnostic and prognostic qualities with positive results [67,86,93]. MMP-7 is not the only MMPs that has shown promising results as a diagnostic biomarker. Recently, Todd et al. evaluated the circulating serum levels of MMPs (MMPs 1, 2, 3, 7, 8, 9, 12, and 13) and tissue inhibitors of MMPs (TIMP) (TIMPs 1, 2, and 4) in a cohort of 300 IPF patients from the IPF-PRO Registry, highlighting that MMPs and TIMPs analyzed were all present at higher levels in patients with IPF compared with controls except for TIMP2. MMP8, MMP9, and TIMP1 were the best diagnostic markers for distinguishing patients with IPF from controls. Moreover, MMP7, MMP12, MMP13, and TIMP4 were able to stratify patients for disease severity [94].

The evidence on the prognostic value of OPN is scarce [72,95]. However, interestingly, a recent study showed that OPN serum levels where significantly higher in patients with AE of IPF compared with stable IPF or healthy controls. Moreover, higher levels of OPN were associated to increased mortality in AEs [96]. Periostin, another ECM protein involved in tissue development and wound healing, has been shown part of the pathogenetic process in IPF. Periostin has prognostic values, in fact total periostin can predict both short-term declines of pulmonary function and overall survival in IPF patients. However, total periostin is not specific for IPF. On the contrary, the monomeric periostin form is more specific and can be used not only to predict pulmonary function decline but also to distinguish IPF patients from healthy controls [97].

## 7. Therapeutic Biomarkers

Therapeutic biomarkers should provide quantitative assessment or indicate the presence or absence of mechanisms or biological pathways targeted by therapy. Since these biomarkers work as surrogate endpoints, they should be measurable over time with low risks for patients, low intra-patient, inter-test variability, and should improve clinical decision making of therapeutic intervention. Finally, when a therapeutic biomarker is evaluated threshold for change should be established reflecting meaningful therapeutic response [14].

Since many biomarkers have been studied before the introduction of antifibrotic therapy, evidence on their usefulness in monitoring response to therapy is more limited.

Surfactant proteins serum levels have shown potential efficacy as outcomes in IPF therapy. SP-A levels in IPF patients treated with pirfenidone or nintedanib from baseline to 3 and 6 months were found to predict progression [98], while SP-D levels in IPF patients treated with IPF predict disease progression and prognosis [99,100]. 

Promising data have been shown in several studies on KL-6 and its use in the monitoring of antifibrotic therapy. Response to pirfenidone therapy correlates with changes in serum KL-6 over time in one study [101] Bergantini et al. evaluated serial measurements of serum KL-6 in IPF patients treated with nintedanib, and demonstrated an indirect correlation with forced vital capacity (FVC) percentages and KL-6 values. Moreover, after 1 year of treatment, patients on therapy showed stable FVC percentages and KL-6 levels compared with baseline values [102]. Nakamura et al. did not find any difference in KL-6 serum values in severe IPF patients treated with nintedanib compared with non-severe patients [103].

MMP-7 has recently been studied in with other biomarkers of transplant free survival in a study on 325 patients, 68 of them treated with antifibrotic therapy, to evaluate the role of such biomarkers in patients on antifibrotic therapy. The study revealed that these biomarkers predict differential transplant free survival in patients on antifibrotic therapy but at higher thresholds than in non-treated patients. Moreover, plasma biomarker level generally increases over time in non-treated patients but remain unchanged in patients on antifibrotics [104].

## 8. Conclusions

The need of reliable biomarkers is becoming more and more fundamental. The validation of useful and accurate diagnostic markers could reduce uncertainty and the use of invasive procedure. Prognostic and therapeutic markers could help stratify patients based on severity and disease behavior in order to personalize management. Moreover, reliable markers able to predict AEs could implement prevention measures and modify the prognosis of such events, which, to date, is poor. Several molecules have shown potential value as biomarkers in IPF. However, many of them have been evaluated mainly in Asiatic cohorts of patients, where their use is more common. Their accuracy should be confirmed also in Caucasian cohorts in order to routinely apply them in the management of IPF. The use of biomarker index composed by multiple biomarkers already studied separately, with the aim of improve diagnostic accuracy in distinguish IPF from other ILDs or healthy controls is promising, but, for now, has shown controversial results.

Finally, none of these biomarkers have been validated in large clinical trials, which still remain an unmet need. However, as a remark of the importance of biomarkers in IPF, many clinical trials evaluating as primary or secondary outcomes known and new biomarkers, have been conducted (Table 2 and Table 3) or are still ongoing (Table 3 and Table 4). 

## Figures and Tables

**Figure 1 ijms-22-06255-f001:**
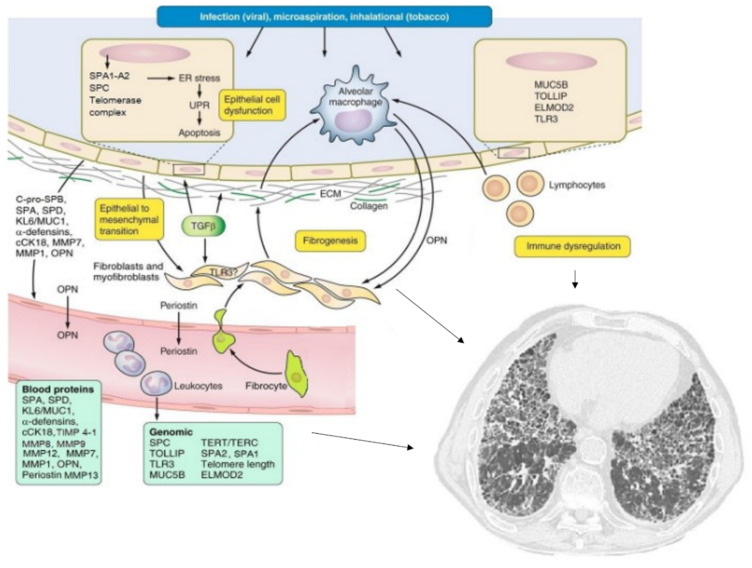
Pathogenesis and molecular biomarkers of idiopathic pulmonary fibrosis. Various mechanisms (most of them indicated by the arrows) leads to pulmonary fibrosis (see text for details). Adapted from Ley B, et al. *Am. J. Physiol. Lung Cell Mol. Physiol*. **2014**, *307*, L681–L691 [14].

**Table 1 ijms-22-06255-t001:** Molecular biomarkers in IPF.

Biomarker	Predisposition	Diagnosis	Prognosis	Therapy Monitoring
SP-C SP-A SP-D C-pro-SP-B	Disease: ++ Disease: ++	Disease: ++ Disease: + AE: ++ Disease: ++	Disease: ++ Disease: ++	Disease: + Disease: +
MUC5B	Disease: +++		Disease: +/−	Disease: ++
Telomerase complex	Disease: +	Disease: ++	Disease: ++	
TLRs	Disease: +	Disease: +	Disease: ++	
ELMOD-2	Disease: +			
KL-6/MUC1		Disease: +	Disease: + AE: ++	Disease: ++
cCK18		Disease: ++	Disease: −	
MMPs:		Diagnosis: ++	Disease: +++	Disease: +++
OPN		Disease: +	Disease: − AE: ++	
TOLLIP			Disease: ++	
α-defensins		Disease: + AE: ++		
Periostin			Disease: ++	

AE: acute exacerbation; SP surfactant protein; MUC5B mucin 5B; TLRs Toll-like receptors; KL-6/MUC1 Krebs von den Lungen-6; cCK18 Circulating caspase-cleaved cytokeratin-18; MMPs metalloproteases; OPN osteopontin.

**Table 2 ijms-22-06255-t002:** Clinical trials on predisposition, diagnostic and prognostic biomarkers in IPF.

		Primary Outcome	Secondary Outcomes	Biomarkers Considered	Type of Biomarker	Status and Results
Biomarker Discovery for Novel Drug Development in IPF NCT01718990 Year: 2012	Type: observational prospective longitudinal cohort trial N. part: 110 Patients with IPF vs. healthy volunteers	Dose on BAL, alveolar macrophages, and blood of mechanistically informative markers of alveolar epithelial cell ER stress, αvβ6-mediated TGFβ activation, and EMT	/	Mechanistically informative markers	Diagnostic Therapeutic	Status: completed Results:/
Exhaled Breath Condensate Biomarkers and Cough in People with IPF NCT02630940 Year: 2015	Type: cross-sectional cohort study N. part: 52 IPF cohort	Detection of 8-isoprostane levels in patients’ exhaled breath condensate samples	LCQ KBILD MRC dyspnoea scale Visual analogue scale for Cough Non-validated acceptability questionnaire	8-isoprostane in exhaled breath condensate	Prognostic	Status: completed Results:/
Prospective Evaluation of Biomarker Profiles in IPF NCT02151435 Year: 2014	Type: Observational perspective N. part: 43 IPF cohort	Progression-free survival at 1 year	Longitudinal change in biomarker levels	Peripheral blood biomarkers based on extracellular matrix and matrix-modifying molecules	Prognostic	Status: completed Results:/
COMET study NCT01071707 Year: 2010	Type: Observational perspective N. part: 108 IPF cohort	Progression free survival as determined by time until any of: death, AE of IPF, relative change in FVC (liters) of at least 10% or DLCO (ml/min/mmHg) of 15% (min 16 weeks; max 80 weeks FU)	/	Multiple biomarkers at baseline (from blood, BAL, bioptic lung tissue)	Prognostic	Status: completed Results: - Progression of IPF is associated with the presence of specific members within the Staphylococcus and Streptococcus genera. Disease progression was significantly associated with increased two OTUs-Streptococcus OTU 1345 (relative risk 1.11, 95% CI 1.04–1.18; *p* = 0.0009) and Staphylococcus OTU 1348 (1.16, 1.03–1.31, *p* = 0.012). DOI: 10.1016/S2213-2600[14]70069-4 - Serial transcriptomic change predicts future FVC decline. Analysis of cell types involved in the progressor signature supports the novel involvement of NK cells in IPF progression. DOI: 10.1164/rccm.202008-3093OC)
PROFILE—Central England NCT01134822 Year: 2010	Type: observational prospective N. part: 330 IPF/NSIP cohort	Discover biomarkers in IPF (discover and validate novel biomarkers, prospectively validate a panel of previously published biomarkers, investigate genetic associations and epigenetic modifications which affect disease severity and progression)	Survival from pulmonary fibrosis (up to 10 years)	Multiple biomarkers	Diagnostic Prognostic	Status: completed Results: - Serum biomarkers (SP-D, MMP-7, CA19-9, and CA-125) can be used to predict disease progression and death in IPF. Surfactant protein D (46.6 ng/mL vs. 34.6 ng/mL, *p* = 0·0018) and CA19-9 (53.7 U/mL vs. 22.2 U/mL; *p* < 0.0001) were significantly higher in patients with progressive disease than in patients with stable disease, and rising concentrations of CA-125 over 3 months were associated with increased risk of mortality (HR 2.542, 95% CI 1.493–4.328, *p* = 0.00059). DOI: 10.1016/S2213-2600[17]30430-7 - Concentrations of protein fragments generated by MMP activity are increased in the serum of individuals with IPF compared with healthy controls. Mean concentrations of C1M (*p* = 0.001), C3M (*p* = 0.044), C6M (*p* = 0.003), and CRPM (*p* = 0.024) at baseline were higher in patients with IPF than in healthy controls. When assessed longitudinally, concentrations of six neoepitopes (BGM, C1M, C3A, C3M, C6M, and CRPM) were significantly higher in patients with progressive IPF than in patients with stable idiopathic pulmonary fibrosis by 6 months. Baseline concentrations of two neoepitopes were associated with increased mortality (C1M: HR 1.62 (95% CI 1.14–2.31), *p* = 0·0069; C3A: 1·91 [1.06–3.46], *p* = 0.032). The rate of change between baseline and 3 months of six neoepitopes (BGM: HR 1.084 [95% CI 1.03–1.14], *p* = 0.0019; C1M: 1.01 [1.003–1.017], *p* = 0.0039; C3M: 1.106 [1.045–1.170], *p* = 0.0005; C5M: 1.003 [1.001–1.005], *p* = 0.0011; C6M: 1.042 [1.007–1.078], *p* = 0.017; and CRPM: 1.38 [1.16–1.63], *p* = 0.0002) was strongly predictive of overall survival, and the increased risk was proportional to the magnitude of change in neoepitope concentrations. DOI: 10.1016/S2213-2600[15]00048-X
PROFILE_Brompton Study NCT01110694 Year: 2010	Type: observational prospective N. part: 230 IPF/NSIP cohort	Discover and validate novel biomarkers and gene expression profiles for use in subsequent clinical studies in patients with IPF.	Prospectively evaluate longitudinal disease behavior in patients with IPF and other fibrotic lung diseases with a view to developing composite clinical endpoints for subsequent use in clinical studies in patients with pulmonary fibrosis. Identify differences in the pathogenetic mechanisms involved in the development of different types of fibrosis	Multiple biomarkers	Diagnostic Prognostic	Status: completed Results: as above
It’s Not JUST IPF Study NCT03670576 Year: 2018	Type: observational prospective N. part: 250 Fibrotic Lung disease cohort (4 categories: RA-UIP, Asbestosis, Chronic HP and Unclassifiable) vs. IPF	Disease progression defined as >10% relative decline in FVC Overall survival	Serum and Plasma Biomarkers (SP-D, MUC16, CA199, Nordic Neoepitopes), DLCO and QoL at 3,6,12 and 24 months Domiciliary spirometry	Plasma Biomarkers (SP-D, MUC16, CA199, Nordic Neoepitopes)	Prognostic	Status: suspended (due to COVID-19 pandemics) Results:/
Exhaled Breath Analysis by Secondary Electrospray Ionization—Mass Spectrometry (SESI-MS) in patients with IPF NCT02437448 Year: 2015	Type: prospective observational N. part: 40 20 IPF patients vs. 20 healthy controls	IPF specific mass spectrometric profile of volatile organic compounds of exhaled breath analysis (markers of IPF in exhaled breath)	/	Amino acids	Predisposition	Status: completed Results: exhaled breath of IPF patients showed higher levels of proline, 4-hydroxyproline, alanine, valine, leucine/isoleucine and allysine compared with healthy controls (*p* < 0.05)
IPFJES NCT03211507 Year: 2017	Type: observational (case-control) prospective N. part: 960 IPF males vs. male controls	Association between asbestos exposure and IPF	Dose-response relationship between asbestos exposure and IPF Gene-environment interaction (for MUC5 B rs35705950 and asbestos exposure) odds ratio	MUC5B rs35705950	Predisposition	Status: completed Results:/
Microarray Analysis of Gene Expression in IPF (MAA) NCT00258544 Year: 2005	Type: observational (cohort) N. prat: 80	Identification of genetic markers of IPF	/	/	Predisposition	Status: actrive, not recruiting Results: Eighteen microRNAs including let-7d were significantly decreased in IPF (*p* < 0.05). The down-regulation of let-7d in IPF and the profibrotic effects of this down-regulation in vitro and in vivo suggest a key regulatory role for this microRNA in preventing lung fibrosis. DOI: 10.1164/rccm.200911-1698OC. Epub 2010 Apr 15.
Study to investigate longitudinal changes in breath biomarkers in IPF VOC (BI 1199-0311) ISRCTN18106574 Year: 2018	Type: observational longitudinal cohort N. part: 88	VOC, measured using mass spectrometry, that can distinguish between IPF patients based on their baseline GAP stage (I, II or III)	VOC, measured using mass spectrometry, that can distinguish between patients based on change in FVC after 12 months VOC which can distinguish between patients with an increase in MRC dyspnoea score of 1 or more after 12 months and those without a change VOC that can distinguish between patients with an increase in USCD, SOBQ scores of 5 or more after 12 months compared to those without a change VOC that can distinguish between patients that respond to antifibrotic treatments and those that do not VOC that can distinguish between patients having an AE of IPF and those who are not	Volatile Organic Compounds	Prognostic	Status: completed Results:/

RCT: randomized controlled trial; IPF idiopathic pulmonary fibrosis; CRMP C-reactive protein degraded by matrix metalloproteinase-1/8;; BAL broncho alveolar lavage; FVC forced vital capacity; DLCO diffusion capacity for carbon monoxide; ER endoplasmic reticulum; TGFβ transforming growth factor β; EMT epithelial-mesenchymal transition; FU follow-up; LCQ Leicester Cough Questionnaire; MRC medical research council; NK natural killer; CA19-9 Carbohydrate Antigen 19-9;CA125 Carbohydrate Antigen 125;MUC16 mucin 16; MUC5B mucin 5B; CA199 Carbohydrate Antigen 199; KBILD Kings brief interstitial lung disease questionnaire; NSIP non-specific interstitial pneumonia; AE acute exacerbation; SP-D surfactant protein D; MMP-7 matrix metalloprotease 7; RA-UIP rheumatoid arthritis UIP; HP hypersensitivity pneumonia; QoL quality of life; BALF broncho alveolar lavage fluid;; VOC Volatile Organic Compounds; GAP Gender, Age, and Physiology score; USCD University of California San Diego Shortness of Breath questionnaire; SOBQ shortness of breath questionnaire; OTU operational taxonomic unit.

**Table 3 ijms-22-06255-t003:** Therapeutic biomarkers in IPF.

		Primary Outcome	Secondary Outcomes	Biomarkers Considered	Status and Results
INMARK study NCT02788474 Year: 2016	Type: RCT N. part: 347 Nintedanib vs. Placebo	The rate of change (slope) in blood CRPM from baseline to week 12.	Percentage of patients with disease progression The rate of change in blood C1M from baseline to week 12 The rate of change in blood C3M from baseline to week 12	CRPM C3M C1M	Status: completed Results: rate of change in CRPM is not a marker of response to nintedanib in patients with IPF The rate of change in CRPM from baseline to week 12 was −2.57 × 10^−3^ ng/mL/month in the nintedanib group and −1.90 × 10^−3^ ng/mL/month in the placebo group (between-group difference −0.66 × 10^−3^ ng/mL/month [95% CI −6.21 × 10^−3^ to 4.88 × 10^−3^]; *p* = 0.8146).
A Randomized, Double-blind, Placebo-controlled, Crossover Study to Assess the Effect of 28 Day Treatment with Fostair^®^ Pressurized Metered-dose Inhaler (pMDI) 200/12 on Biomarkers of Platelet Adhesion in Patients with IPF NCT02048644 Year: 2014	Type: RCT N. part 20 beclomethasone/formoterol pMDI 100/6 mcg 2 puffs twice daily for 28 days vs. placebo	Platelet-monocyte complex formation platelet P-selectin expression platelet fibrinogen binding	FVC sputum eosinophils cells six minutes-walk d istance	Platelet derived markers	Status: completed Results: Change from baseline spirometric measurements of FEV1(L), FEV/FVC % pred FEF25–75 were significantly improved following 28 days B/F by (mean ± SD), 0.88 ± 0.16 L (*p* = 0.03), 0.03 ± 0.03 (*p* = 0.03), 12.4 ± 19.1% (*p* = 0.02) respectively when compared to placebo. There was no change in quality of life or exercise measures. The effects of beclomethasone/formoterol in this study may represent delivery of corticosteroid to the peripheral airways ameliorating local injury and altering platelet activation
Randomized, Double-Blind, Placebo-Controlled, Multiple Dose, Dose-Escalation Study of STX-100 in Patients With IPF NCT01371305 Year: 2011	Type: RCT N. part: 41 SXT-100 vs. placebo in IPF	Number of Participants with adverse events	Pharmacodinamic and pharmacokinetic parameters of BG00011 (STX-100) Percentage change from baseline in biomarkers solated from BAL Number of participants with treatment emergent antibodies to BG00011	The expression level of 7 genes; ALOX5, FN1, OLR1, PAI-1 (aka SERPINE 1), TGM2, TREM1, and ETS1 were assessed via BAL as well as a ratio of pSMAD2 to tSMAD2 levels.	Status: completed Results:/
A Open-label, Multicenter Study, With a Single Intravenous Dose of QAX576 to Determine IL-13 Production in patients with IPF NCT00532233 Year: 2007	Type: open label clinical trial N. part: 52 QAX576 in IPF cohort	To investigate the possibility that some IPF patients experience increased IL-13 production. Blood samples to be collected pre-dose and weekly after dosing. -To investigate the hypothesis that QAX576 will neutralize IL-13 in patients with IPF	To evaluate the changes in biomarkers in blood over time in patients with IPF. Serum samples will be obtained at pre-dose and 2 weeks post-dose.	IL-13 Other blood biomarkers	Status: completed Results: the study was terminated early after 31 patients were enrolled and randomized to receive QAX576 due to slow enrolment rate.
Randomized, Double-Blind, Parallel Group, Placebo-Controlled, Multicenter, Exploratory Phase IIa Study to Assess Safety, Tolerability, Pharmacokinetic and Pharmacodynamic Properties of GLPG1690 Administered for 12 Weeks in Subjects With IPF NCT02738801 Year: 2016	Type: RCT N. part: 23 GLPG1690 capsules, administered at a dose of 600 mg, orally QD vs. placebo in IPF cohort.	Adverse events, pharmacodynamic and pharmacokinetics parameters, mean Peak Area Ratio of LPA C18:2 species in Blood and BALF	/	LPA C18:2	Status: completed Results: concentrations of LPA C18:2 in plasma decreased after administration of GLPG1690 at the week 4 (*p* = 0.0001) and 12 (*p* = 0.0014) visits and return to baseline concentrations at the FU visit. In BALF, LPA C18:2 and LPA C20:4 concentrations are below the level of quantification for more than 25% of baseline samples obtained from patients in the GLPG1690 treatment group.
A Randomized, Double-Blind, Placebo-Controlled Phase II Clinical Trial of GKT137831 in Patients with IPF NCT03865927 Year: 2020	Type: RCT N. part: 60 GKT137831 400 mg bid for 24 weeks vs. placebo	Surrogate biomarker of oxidative stress by mass spectroscopy through 24 weeks (changes in concentrations of circulating o,o’-dityrosine)	Collagen degradation product (serum C1M) by enzyme linked immunoabsorbant assay through 24 weeks LFT Ambulatory ability by measuring walk distance in six-minutes Evaluation of safety by adverse events	o,o’-dityrosine C1M	Status: ongoing
Non-Interventional Collecting Evidences For ILD in Taiwan: Optimized Novel Therapy NCT04614441 Year: 2020	Type: observational prospective N. part: 500 IPF vs. PF-ILD vs. SSc-ILD on therapy with Nintedanib 150 mg bid	Annual percentage of decline from baseline in FVC, %, DLCO, % and resting and exercise oxygen saturation (SpO2, %) per cohort of IPF, SSc-ILD, or PF-ILD	Time to first AE of IPF; or time to ILD worsening for SSc-ILD/PF-ILD after study enrolment Annual change from baseline in SGRQ for IPF or K-BILD for other ILDs, CAT, Berlin questionnaire and 6MWT Change from baseline in quantification of biomarkers Mortality	Include but not limited to PDGF, VEGF, FGF, TGF-β1, HGF, MMPs: MMP-1, MMP-7, MMP-9, α-defensin 1, HMGB1, TIMP, HSP: HSP-27, bile acid conjugated, LPA, LPAR1, PGE2, IL: IL-1β, IL-4, IL-18, IL-13, IL-17, MCP-1, MIP-2, periostin, osteopontin, SP-A, SP-D, KL-6/MUC1, anti-HSP70, IgG BMP, CA-199, CRPM, CCL 2, CCL-18	Status: ongoing
Targeted Removal of Pro-Inflammatory Cells: An Open Label Human Pilot Study in IPF NCT02874989 Year: 2016	Type: RCT N. part: 26 Dasatinib + Quercetin vs. placebo in IPF cohort	Percentage of pro-inflammatory expressing cells (skin biopsy) Percentage of pro-inflammatory expressing cells (skin biopsy) BP, weight, HR, CBC, lipid panel, HBA1c, comprehensive metabolic panel, high sensitivity CRP, plasma IL-6, plasma PASP biomarkers, p16INK4a biomarker	/	high sensitivity CRP, plasma IL-6, plasma PASP biomarkers, p16INK4a biomarker	Status: ongoing
EXCHANGE-IPF NCT03584802 Year: 2018	Type: RCT N. part: 40 Therapeutic plasma exchanges vs. conventional treatment in AE of IPF	Overall mortality at day 28 after initiation of therapy	[…] Changes in lung injury biomarkers in plasma (KL-6, SP-D) between day 1 and day 90 Changes in circulating autoantibodies levels (anti-periplakin, anti-HSP70 and anti-vimentin antibodies) between day 1 and day 90	Injury biomarkers Circulating fibrocytes Auto-antibodies	Status: ongoing

RCT: randomized controlled trial; CRPM, C-reactive protein degraded by matrix metalloproteinase; pMDI, pressurized metered dose inhaler; FVC, forced vital capacity; FEV1, Forced Expiratory Volume in the 1st second; FEF, Forced mid-expiratory flow rate; SD, standard deviation; IPF, idiopathic pulmonary fibrosis; BAL, broncho alveolar lavage; IL-13, interleukin-13; C1M, Collagen 1 Degraded by Matrix Metalloproteinase-2/9/13; C3M, Collagen 3 Degraded by Matrix Metalloproteinase-9; ALOX5, Arachidonate 5-lipoxygenase; FN1, fibronectin 1; OLR1, Oxidized low density lipoprotein receptor 1; PAI-1, Plasminogen activator inhibitor-1; TGM2, Transglutaminase 2; TREM 1, Triggering receptor expressed on myeloid cells 1; ETS1, v-ets erythroblastosis virus E26 oncogene homolog 1; LPA, Lysophosphatidic Acid; BALF, broncho alveolar lavage fluid; LFT, lung function test; PF ILD, progressive fibrosing interstitial lung disease; Ssc ILD, systemic sclerosis interstitial lung disease; DLCO, diffusing capacity for carbon monoxide; AE, acute exacerbation; ILD, interstitial lung disease; SGRQ, Saint George Respiratory questionnaire; K-BILD, King’s Brief Interstitial Lung Disease questionnaire; CAT, Chronic obstructive pulmonary disease Assessment Test; 6MWT, 6 min walking test; PDGF, Platelet Derived Growth Factor; VEGF, Vascular Endothelial Growth Factor; FGF, Fibroblast Growth Factor; TGF-β1, Transforming Growth Factor β1; HGF, Hepatocyte Growth Factor; MMPs, metalloproteases; HMGB1, High Mobility Group Box 1; TIMP, Tissue of Metalloproteinase; HSP, Heat-Shock Protein; LPA, Lysophosphatidic Acid; LPAR1, Lysophosphatidic Acid Receptor 1; PGE2, Prostagladin E2; IL, Interleukin; MCP-1, Monocyte Chemoattractant Protein 1; MIP-2, Macrophage Inflammatory Protein 2; SP, surfactant protein; KL6/MUC1, Krebs von den Lungen-6; IgG, Immunoglobolin G; BMP, Bone Morphogenic Protein; CA-199, Carbonhydrate Antigen-199; CRMP, C-reactive protein degraded by matrix metalloproteinase-1/8; CCL, chemokine (C-C motif) ligand; BP, blood pressure; HR, heart rate; CBC, complete blood count; HBA1c, Hemoglobin A1c; CRP, C reactive protein; PASP, pulmonary artery systolic pressure.

**Table 4 ijms-22-06255-t004:** Ongoing clinical studies on biomarkers in IPF.

		Primary Outcome	Secondary Outcomes	Biomarkers Considered	Type of Biomarker
Early Diagnosis of Pulmonary Fibrosis—Use of Biomarkers in IPF NCT02755441 Year: 2016	Type: observational perspective N. part: 300 IPF cohort	Disease progression or mortality at 1 year	Hospitalizations Exacerbations LFTs Mortality QoL Combined endpoints of disease progression Progression in serum/plasma biomarker levels	Unspecified multiple biomarkers	Prognostic
Immunopathologic Profiles of the Lung Micro-Environment Using Cryobiopsies and Identification of Blood Biomarkers in Patients With IPF NCT04187079 Year: 2017	Type: observational prospective N. part: 100 IPF cohort vs. other ILD cohort	Expression of PD-L1 in the epithelial cells in lungs	/	PD-L1, PD-L2, Beta- catenin, B-cell follicles and Tenascin- C in cryobiopsies from the lungs anti HSP 70, p-ANCA, c-ANCA, CD4+/CD28- and CD8+/CD28- cells in blood samples	Diagnostic
Development of Airway Absorption Sampling Methods for Biomarker Assessment in Probable IPF Patients NCT04494334 Year: 2020	Type: observational cross-sectional study N. part: 60 IPF vs. sarcoidosis vs. healthy controls	Levels of the of biomarker/mediator SP-D, CCL18, CXCL13 and periostin in bronchial Lining fluid in IPF and sarcoidosis patients	Levels of Periostin, SP-D, CCL18 and CXCL13 in nasosorption samples within and across the 3 groups of participants Levels of Periostin, SPD, CCL18 and CXCL13 in blood within and across the 3 groups of participants	SP-D, CCL18, CXCL13 and periostin	Diagnostic
Pulmonary Fibrosis Biomarkers During Exacerbation N CT04442711 Year: 2020	Type: observational prospective N. part: 50 IPF cohort	Mortality at 30 and 90 days	Biomarkers level, change in oxygen need, QoL, need for respiratory support, decline of LFTs at 30 days. Treatment during and after hospitalization	Multiple biomarkers on blood serum and plasma collected within 24 h of hospital admission	Diagnostic Prognostic
LOCK-IPF NCT04268485 Year: 2020	Type: observational prospective N. part: 60 IPF cohort	Change in serum KL-6 level between baseline and 12 months	Change in serum KL-6 level between baseline and 3 and 6 months. Correlation of KL-6 and FVC, DLCO, symptoms, response to antifibrotic therapy and GAP stage at 3, 6 and 12 months to baseline Correlation between KL-6 levels and CPI Difference in KL-6 levels between patients with indeterminate, probable and definite UIP on HRCT	KL-6 on blood	Prognostic
Cardiovascular fibrosis in IPF NCT04177251 Year: 2019	Type: observational case-control prospective study N. part: 168 IPF cohort vs. healthy controls	Presence of cardiac fibrosis in a population of patients with overt IPF at diagnosis in comparison with healthy controls Presence of vascular fibrosis in a population of patients with overt IPF at diagnosis in comparison with healthy controls	Levels of biomarkers analyzed (galectins-3, osteopontin and periostin) IPF progression after 1 year from diagnosis in IPF patients Blood proteomic and metabolomic biomarkers	galectins-3, osteopontin and periostin Proteomic and metabolomic biomarkers	Diagnostic Prognostic
The Role of the miR200 Family in the Restoration of Normal Lung Homeostasis and Detection of Early IPF NCT03457935 Year: 2018	Type: observational prospective N. part: 450 IPF vs. non-IPF ILD vs. healthy controls	Determine miR200 levels (fold change) in blood samples to identify biomarkers for IPF	/	miR200	Diagnostic
IPF and Serum Bank NCT04016168 Year: 2014	Type: observational prospective N. part: 500 Diffuse idiopathic ILD cohort	Determination of circulating CD163 serum concentration	/	CD163	n/a
Role of Genetics in IPF NCT01088217 Year: 2010	Type: observational cross-sectional study (family based) N. part: 8000 IPF, familial pulmonary fibrosis cohort, Idiopathic Interstitial Pneumonia Familial Interstitial Pneumonia	Identify a group of genetic loci that play a role in the development of familial interstitial pneumonia and idiopathic interstitial pneumonia.	Develop biomarkers using proteomic and genomic approaches that will facilitate establishing the diagnosis and prognosis of both familial and sporadic forms of idiopathic interstitial pneumonia	Multiple biomarkers	Diagnostic Prognostic
ELFMEN Study NCT04016181 Year: 2007	Type: observational prosepective N. part: 800 IPF and other ILDs	Time to death	Biomarkers that are associated with increased rate of decline in vital capacity, increased lung-related mortality and that predict rate of change in gas transfer	Multiple biomarkers	Prognostic
Genomic and Proteomic Analysis (GAP) of Disease Progression in IPF NCT00373841 Year: 2006	Type: observational N. part: 500 IPF cohort	Identify genetic and biologic markers that may predict the loss of lung function due to idiopathic pulmonary fibrosis through comparison of genetic and biologic markers of samples to changes in symptoms	/	Multiple biomarkers	Prognostic
EXCHANGE-IPF NCT03584802 Year: 2018	Type: RCT N. part: 40 Therapeutic plasma exchanges vs. conventional treatment in AE of IPF	Overall mortality at day 28 after initiation of therapy	[…] Changes in lung injury biomarkers in plasma (KL-6, SP-D) between day 1 and day 90 Changes in circulating autoantibodies levels (anti-periplakin, anti-HSP70 and anti-vimentin antibodies) between day 1 and day 90	Injury biomarkers Circulating fibrocytes Auto-antibodies	Therapeutic

RCT: randomized controlled trial; IPF, idiopathic pulmonary fibrosis; ILD, interstitial lung disease; LFTs, lung function tests; QoL, quality of life; PD-L1/2, Programmed Death-Ligand ½; HSP 70, heat shock protein 70; ANCA, Antineutrophil Cytoplasmic Antibodies; CD 4-28-8-163, cluster of differentiation 4-28-8-163; SP-D/A, surfactant protein D/A; CCL18/2, chemokine ligand 18/2; CXCL13, CXC motif chemokine 13; KL-6/MUC1, Krebs von den Lungen 6/Mucin 1; FVC, forced vital capacity; DLCO, diffusion capacity for carbon monoxide; GAP, Gender, Age, and Physiology score; UIP, usual interstitial pneumonia; HRCT, high resolution computed tomography; C1M, Collagen 1 Degraded by Matrix Metalloproteinase-2/9/13.

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
