# Peer review of "Molecular Biomarkers in Idiopathic Pulmonary Fibrosis: State of the Art and Future Directions"

_ijms, 2021, doi:10.3390/ijms22126255_

Round 1

Reviewer 1 Report

The authors have done significant and very interesting work to systematize clinical trials.

1) However, it seems to me that a more detailed presentation of the results of clinical trials (including an indication of the statistical significance and power of statistical hypotheses) would be more informative.

2) In addition, it may be worth dividing the research into several groups. In particular, studies related to the use of drugs should be reasonably separated from studies of the pathogenesis of the disease.

3) It would also be useful to conduct a comparative analysis of the diagnostic and prognostic reliability of various markers.

Author Response

Reviewer 1

 R1.C1

The authors have one significant and very interesting work to systematize clinical trials.

R1.R1

We thank the Reviewer for appreciating our manuscript.

R1.C2

However, it seems to me that a more detailed presentation of the results of clinical trials (including an indication of the statistical significance and power of statistical hypotheses) would be more informative.

R1.R2

We thank the Reviewer for the comment. When they were available, we updated the results of the trials more in detail, as you can in table 2 and table 3.

 R1.C3

In addition, it may be worth dividing the research into several groups. In particular, studies related to the use of drugs should be reasonably separated from studies of the pathogenesis of the disease.

R1.R3

We thank the Reviewer for raising this important point. We added a new table (table 3) separating biomarkers targeting specific drugs from those diagnostic, prognostic and predisposing disease development.

R1.C4

It would also be useful to conduct a comparative analysis of the diagnostic and prognostic reliability of various markers.

R1.R4

We thank the Reviewer for his/her comment. A comparative table of the biomarkers included in the review has been added to summarize reliability of various markers (table 1).

Reviewer 2 Report

The authors present a well-organised and coherent review concerning the molecular biomarkers in IPF. Despite the broad spectrum of the theme, the authors manage to review the recent litterature regarding the most significant biomarkers in a comprehensive way. I would only suugest to include a figure summarizing their key findings about the core mechanisms and candidate molecular biomarkers for idiopathic pulmonary fibrosis.

Author Response

R2.C1

The authors present a well-organised and coherent review concerning the molecular biomarkers in IPF. Despite the broad spectrum of the theme, the authors manage to review the recent litterature regarding the most significant biomarkers in a comprehensive way. I would only suugest to include a figure summarizing their key findings about the core mechanisms and candidate molecular biomarkers for idiopathic pulmonary fibrosis.

R2.R1

We thank the Reviewer for the comment. A table summarizing the biomarkers included in the review has been added.